# Impact of natural disasters on HIV risk behaviors, seroprevalence, and virological supression in a hyperendemic fishing village in Uganda

Hadijja Nakawooya[1]*, Victor Ssempijja[1,2], Anthony Ndyanabo[1], Ping Teresa Yeh[3], Larry W. Chang[1,3,4,5], Maria J. Wawer[1,5], Fred Nalugoda[1], David Serwadda[1,6], Ronald H. Gray[1,5], Joseph Kagaayi[1,6], Steven J. Reynolds[1,6,7], Tom Lutalo[1,8], Godfrey Kigozi[1], M. Kate Grabowski[1,9‡], Robert Ssekubugu[1‡]

1 Rakai Health Sciences Program, Kalisizo, Uganda, 2 Clinical Monitoring Research Program Directorate, Frederick National Laboratory for Cancer Research, Frederick, MD, United States of America, 3 Department of International Health, Johns Hopkins Bloomberg School of Public Health, Baltimore, MD, United States of America, 4 Department of Medicine, Johns Hopkins School of Medicine, Baltimore, MD, United States of America, 5 Department of Epidemiology, Johns Hopkins Bloomberg School of Public Health, Baltimore, MD, United States of America, 6 Makerere University School of Public Health, Kampala, Uganda, 7 Division of Intramural Research, National Institute of Allergy and Infectious Diseases, National Institutes of Health, Bethesda, MD, United States of America, 8 Uganda Virus Research Institute, Entebbe, Uganda, 9 Department of Pathology, Johns Hopkins School of Medicine, Baltimore, MD, United States of America

‡ MKG and RS are contributed equally to this work as co-senior authors.
* hnakawooya@rhsp.org

**Data Availability Statement:** We appreciate the effort that the PLOS journal family takes to make data available (and to guarantee long-term stability

## Abstract

### Background

Understanding the impact of natural disasters on the HIV epidemic in populations with high HIV burden is critical for the effective delivery of HIV control efforts. We assessed HIV risk behaviors, seroprevalence, and viral suppression in a high HIV prevalence Lake Victoria fishing community before and after COVID-19 emergence and lockdown and a severe lake flooding event, both of which occurred in 2020.

### Methods

We used data from the largest Lake Victoria fishing community in the Rakai Community Cohort Study, an open population-based HIV surveillance cohort in south-central Uganda. The data were collected both prior to (September-December 2018) and after (October-December 2021) COVID-19 emergence and a severe flooding event. Households impacted by flooding were identified via drone data and through consulting village community health workers. The entire study population was subject to extensive COVID-19-related lockdowns in the first half of 2020. Differences in HIV-related outcomes before and after COVID, and between residents of flooded and non-flooded households, were assessed using a difference-in-differences statistical modeling approach.

and availability of data despite potential changes to the institutional affiliations etc of individual authors). Data requests may be sent to the Rakai Health Sciences Program data management office (datarequests@rhsp.org), where data are archived across all the various projects run by the RHSP (original paper forms from the RCCS surveys, as well as the electronic datasets for each survey round).

**Funding:** The Rakai Community Cohort Study rounds 19 and 20 were supported by the National Institute of Mental Health (R01MH099733, R01MH105313, R01MH107275, R01MH115799), National Institute of Allergy and Infectious Diseases (R01AI114438, K25AI114461, R01AI123002, K01AI125086, R01AI128779, R01AI143333, R21AI145682, R01AI155080), National Institute on Alcohol Abuse and Alcoholism (K01AA024068), Eunice Kennedy Shriver National Institute of Child Health and Human Development (R01HD091003), National Heart, Lung, and Blood Institute (R01HL152813), and the Bill and Melinda Gates Foundation (OPP1175094). The study was also supported in part by the Division of Intramural Research, National Institute of Allergy and Infectious Diseases. HN received training and support from National Institutes of Health Fogarty International Center (D43TW010557). The funders were not involved in the design of the study and collection, analysis, and interpretation of data or in writing the manuscript.

**Competing interests:** The authors declare no competing interests.

## Findings

A total of 1,226 people participated in the pre- and post-COVID surveys, of whom 506 (41%) were affected by flooding. HIV seroprevalence in the initial period was 37% in flooded and 36.8% in non-flooded households. After the COVID-19 pandemic and lockdown, we observed a decline in HIV-associated risk behaviors: transactional sex declined from 29.4% to 24.8% (p = 0.011), and inconsistent condom use with non-marital partners declined from 41.6% to 37% (p = 0.021). ART coverage increased from 91.6% to 97.2% (p<0.001). There was 17% decline in transactional sex (aPR = 0.83, 95% CI: 0.75–0.92) and 28% decline in the overall HIV risk score (aPR = 0.83, 95% CI: 0.75–0.92) among HIV-seronegative participants. We observed no statistically significant differences in changes of HIV risk behavior, seroprevalence, or viral suppression outcomes when comparing those affected by floods to those not affected by floods, in the periods before and after COVID-19, based on difference-in-differences analyses.

## Interpretation

Despite a high background burden of HIV, the COVID-19 pandemic, and severe flooding, we observed no adverse impact on HIV risk behaviors, seroprevalence, or virologic outcomes. This may be attributed to innovative HIV programming during the period and/or population resilience. Understanding exactly what HIV programs and personal or community-level strategies worked to maintain good public health outcomes despite extreme environmental and pandemic conditions may help improve HIV epidemic control during future natural disaster events.

## Introduction

Globally, natural disasters have increased in frequency and severity, with floods among the most common and devastating disaster events [1, 2]. Over the last decade, African countries have been especially affected by flooding, with climate change driving recent large increases in flooding events [3–5]. The African continent is also home to the largest number of HIV cases globally [6]. Prior research suggests that floods can negatively impact HIV epidemic control due to interruptions in HIV service provision and adaptations in human behaviors [7–9]. Lake Victoria fishing communities in eastern Africa face one of the highest HIV burdens globally, with adult HIV seroprevalence estimates ranging from 20 to 40% and HIV transmission remains high with incidence of 1.59/100 per person years [10, 11]. Many people in these fishing communities engage in high-risk sexual behaviors and drug and alcohol use. Women frequently report engaging in risky sexual behaviors, often related to sex and bar work. The situation is further complicated by increased vulnerability to HIV, including increased peer pressure especially among adolescents, and poor housing comprised of make-shift shelters made of timber often occupied by multiple families of different backgrounds and their children [12].

Residents along the shores of Lake Victoria in East Africa have been experiencing the health, social, and economic impacts of severe flooding events in conjunction with the COVID-19 pandemic in the early 2020s [13–16]. Floods disrupt lives and health practices, raising HIV transmission risks due to changes in behavior and limited prevention resources [9,

17]. Reduced support for individuals with HIV, coupled with limited information on HIV and disruption to counseling and testing services by floods, contribute to undiagnosed cases and unsafe sex practices [18]. Malnutrition and disease further weaken immune systems, increasing likelihood of virus transmission [7, 19]. These factors raise concerns that the hard-earned gains in HIV epidemic control achieved over the previous decade in these high-burden settings may be lost [18, 20, 21].

In Uganda, people living with HIV have experienced extreme stress due to the COVID-19 pandemic. This stress includes worries about access to antiretroviral therapy (ART), concern over inadvertent disclosure of HIV status, fear that coronavirus infection would have more severe outcomes for people living with HIV, and distress due to pandemic-induced poverty and food insecurity [20]. Persons living along the Lake Victoria shoreline experienced additional stresses due to lake flooding events caused by continuous heavy rains [22]. As a result of flooding, communities along the shorelines were grossly affected in terms of their livelihood including displacement of families and households [23]. Despite the extremely high HIV burden in Lake Victoria fishing communities, it remains largely unknown how these dual natural disasters have impacted the state of the HIV epidemic, including possible effects on population HIV-associated risk behaviors and treatment outcomes among people living with HIV.

Using data from the Rakai Community Cohort Study (RCCS), an open longitudinal population-based HIV surveillance cohort, we conducted a pre- and post-study to evaluate the impact of COVID-19 and flooding on the prevalence of HIV-associated risk behaviors and HIV virologic suppression among people living with HIV. Our analysis included data from individuals residing in a large Lake Victoria fishing community that experienced severe flooding in 2020, during the earlier times of COVID-19 pandemic national lockdowns. HIV burden within this community is among the highest in the East African region, with an estimated incidence rate of approximately 1.6 per 100 person-years and HIV seroprevalence of 37% [10]. Specifically, we examined changes in HIV-associated behaviors and treatment outcomes before and after COVID-19 among individuals in this community affected by severe flooding and COVID-19 emergence. Research on health outcomes following natural disasters may help inform public health policies aimed at preventing and reducing negative health outcomes during times of emergency.

## Materials and methods

### Study design and participants

The RCCS is an open population-based longitudinal HIV surveillance cohort in south-central Uganda, implemented in agrarian, semi-urban and four Lake Victoria fishing communities. The RCCS has been ongoing in agrarian and semi-urban trading communities since 1994 and Lake Victoria fishing communities since 2011, and has been described in detail elsewhere [11, 24]. Briefly, eligible participants are identified through a household census enumeration of all person's information on household members' sex, age, and duration of residence obtained regardless of whether they are present in the home at time of census. During census, household geographic coordinates (latitude and longitude) are collected using global positioning system (GPS) technology, using the Garmin eTrex 10 GPS. After the census, the RCCS surveys all residents within the eligible age range (15–49 years) who can provide written informed consent. Participants are then interviewed about their demographic characteristics, sexual behaviors, and ART use. Venous blood is obtained for HIV testing, and viral load assessment is conducted among HIV-positive persons.

This study included RCCS participants residing in the four RCCS fishing communities, including the largest Lake Victoria fishing community in south central Uganda [10]. Analyses

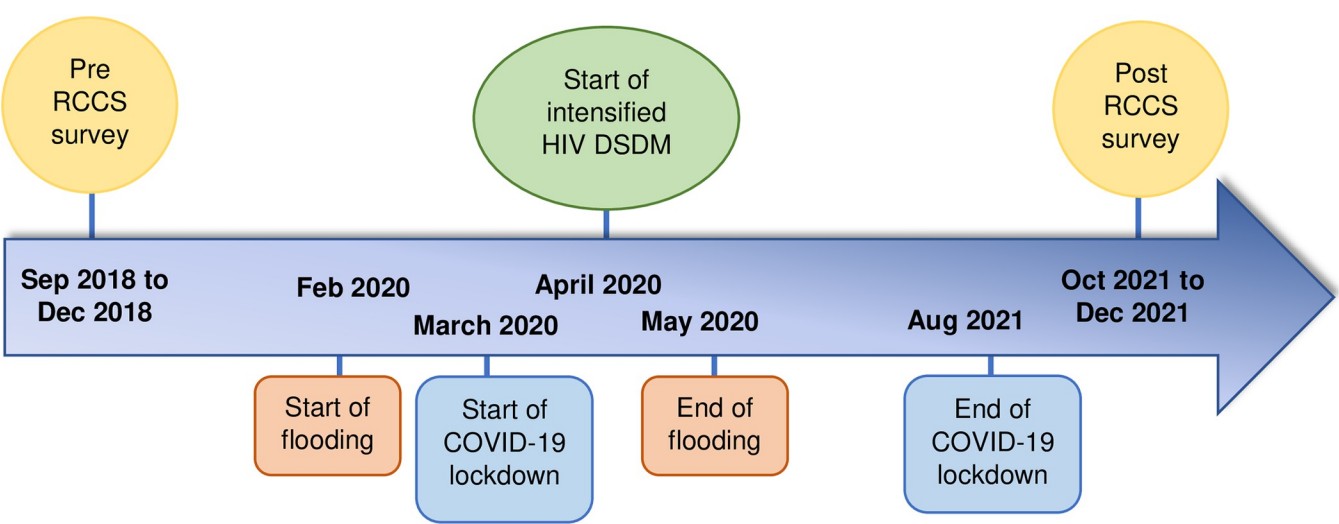

**Fig 1. Timeline of key events.** Events include RCCS survey rounds, Lake Victoria flooding, COVID-19 emergence, and intensified implementation of differentiated HIV service delivery models.

were restricted to two RCCS survey rounds (Fig 1). The pre-study period included the last RCCS survey done prior to the onset of the COVID-19 pandemic in March 2020 and Lake Victoria flooding in February 2020 (conducted between September 3rd, 2018, and December 19th, 2018). The post-study period included the first RCCS survey conducted after the onset of the pandemic and flooding (conducted between October 11th, 2021, and December 16th, 2021). Because of widespread migration during the analysis period, we used longitudinal analysis to focus on individuals who participated in both RCCS surveys and were therefore likely to have been exposed to both emergencies.

## Provision of HIV services in fishing communities during the COVID-19 pandemic and Lake Victoria flooding

The adaptation of HIV services in fishing communities during COVID-19 in Uganda were based on guidelines provided by the Uganda Ministry of Health to all HIV service implementing partners throughout the country. The various models used to accommodate COVID-19 restrictions have been described elsewhere [25]. In general, multiple community-based strategies were used to ensure continued access to essential HIV services during lockdowns and travel restrictions. For example, community drug dispensing points (CDDP) were established: health workers traveled to communities to meet with clients and deliver their medications. Four CDDP sites were operated in this fishing village including one on a nearby island, with at least one site visit every month. In addition, under the community client-led ART delivery model, clients within the same geographical location formed homogenous groups of maximum ten people. In these groups, the group leader (or designated member) would travel to the CDDP or health facility to collect medications for the rest of the members, thereby reducing congestion during drug pickup. Across all the models of differentiated services, multi-month dispensing of medications was encouraged, with stable clients receiving up to six months drug refills. This approach was further supplemented by mobile phone counseling and check-in calls.

## Identification of flooded households

Flooding occurred continuously for six months following heavy rains that started in February 2020 [26]. The floods submerged extended distances of the shores of Lake Victoria, causing

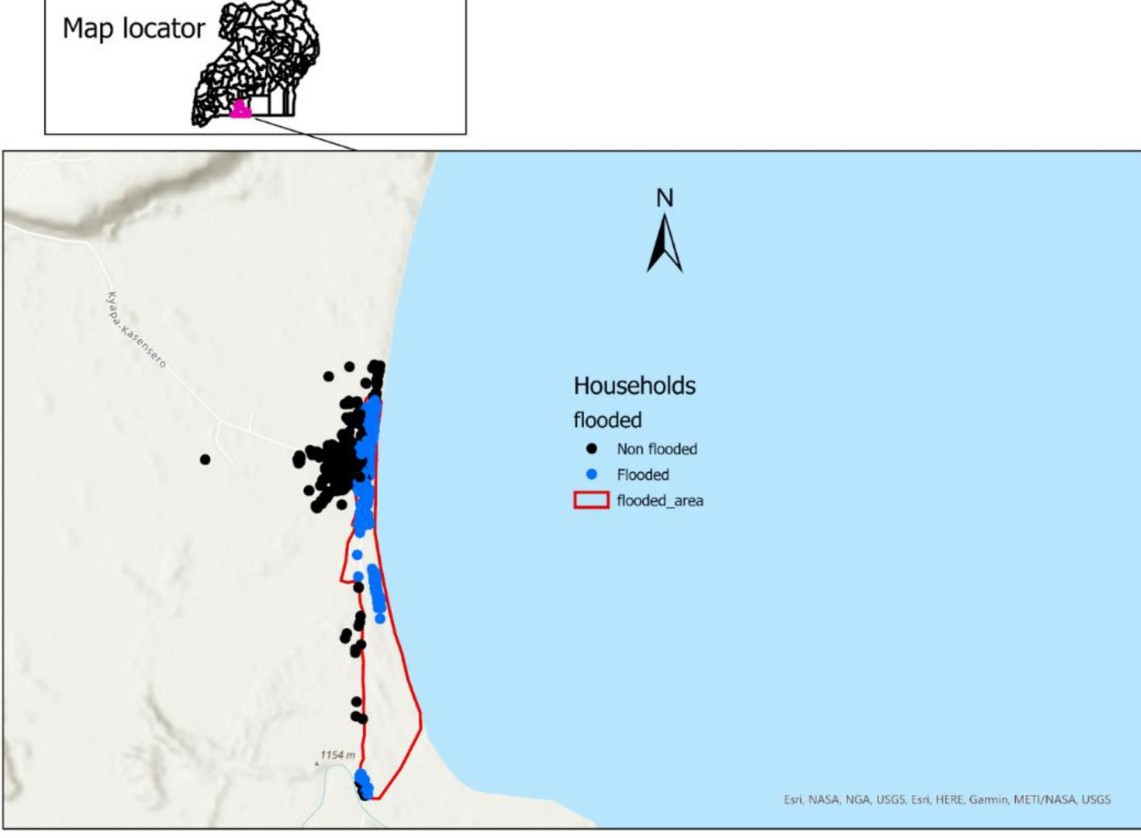

**Fig 2. Map of flooded households following a mass flooding event in spring of 2020 in Kasensero, Uganda.** Map adapted from open-source shapefiles obtained from Stanford University Library's search tool for GIS datasets [27, 28].

some households to be submerged in the water, washed away, or became grossly damaged and uninhabitable. To identify affected households, a drone-supported GPS was launched to map the new shoreline of Lake Victoria after the flooding to determine which households were affected by the floods. By capturing coordinates of the shoreline, we were able to compare these with the previous shoreline to identify which households were located within the flooded area along the lake. This exercise was conducted before the shoreline receded to its original position.

We also identified other offshore patches that were affected by the floods through consultations with village community health workers and local leaders who were present at the time of floodings. After identification of the offshore flooded areas, the drone was used to map and capture coordinates for the offshore line patchy areas as well. We then plotted all active clusters of households that existed before flooding using ARCGIS-PRO and overlaid the current shoreline and the offshore map of the flooded area. All households below the shoreline or in the flooded map were identified as flooded; the rest were identified as non-flooded (Fig 2). We also assessed changes in household location after flooding to identify new areas of settlement.

## Primary outcomes

Primary HIV-associated behaviors and treatment outcomes included number of sexual partners in the past 12 months, inconsistent condom use with non-marital partners, transactional

sex, alcohol use in the past 12 months, HIV prevalence, ART use, and viral suppression. These factors were assessed before and after flooding, using pre-COVID-19 and post-COVID-19 RCCS surveys. HIV risk score among HIV-seronegative participants were measured using a previously validated gender-specific risk index to predict risk of HIV infection [29]. Individuals were assigned a risk score using the following variables: age, marital status, education, number of sexual partners, whether inconsistent condom use occurred, use of alcohol before sex, concurrent sexual partners, men's circumcision status, whether the participant's employment type was associated with high risk of acquiring HIV, and whether partner had a high-risk employment type. ART coverage was defined as the proportion of all participants living with HIV who self-reported ART use. Viral suppression was defined as a viral load less than 1000 copies/ml, in line with WHO recommendations [30]. Each individual reported number of sexual partners in the past 12 months and for those who reported more one sexual partner were categorized as having multiple sexual partners. Transactional sex was defined as self-report of having exchanged money, gifts, or favors for sex.

## Statistical analysis

We first conducted descriptive analyses of demographic variables and key outcomes to characterize the sample overall and assess differences between participants affected by floods before and after emergence of COVID-19 and lake flooding. Direct impacts of household flooding were estimated using difference-in-differences, specifically assessing before-to-after changes in primary outcomes (HIV risk score, ART use, viral load suppression) in flooded and non-flooded households. Modified Poisson regression models were employed for each outcome, incorporating indicators for flooding (whether the household was flooded: yes or no) and survey period (pre- or post-COVID-19) [31, 32]. The difference-in-differences (DID) assessment was conducted by examining the coefficient of the interaction term between these variables, with random effects analysis applied to participant ID to account for the within-subject correlation. Associations were reported were unadjusted and adjusted prevalence ratios (aPR) with 95% confidence intervals (CI). We also conducted a matched-pair-analysis using conditional logistic regression to assess significant changes in primary outcomes over calendar time within individual participants.

## Ethical approval

The RCCS was reviewed and approved by the Research and Ethics Committee at the Uganda Virus Research Institute and it is registered at the Uganda National Council for Science and Technology. Additional approvals were obtained from the Johns Hopkins Committee on Human Subjects Protections. Community consent was obtained through dialogue meetings with local leaders and the community advisory board prior to the start of each survey visit and sometimes intermediately. Verbal informed consent was obtained at household census level from informants or an adult in the household, and both verbal and individual written informed consent was obtained at the RCCS survey participation level by those eligible and willing to participate. All the datasets were anonymized by the Rakai Health Sciences Program data managers. Permission to use drones in the community was obtained from the local leaders and the Uganda Police.

## Results

In the pre-COVID survey, a total of 4,366 were survey-eligible, of whom 2,209 participated (50.6%). The primary reason for non-participation was absence for work or school at time of survey, rather than refusal (S1 Table). Between the pre- and post-COVID-19 surveys, 41.1%

**Table 1. Sociodemographic characteristics of participants by flooding category in the pre- and post- COVID-19 surveys.**

| Demographic characteristics | Pre-COVID | | | Post-COVID | | |
| --- | --- | --- | --- | --- | --- | --- |
| | Non-Flooded (N = 720) | Flooded (N = 506) | Chi-square p-value | Non-Flooded (N = 720) | Flooded (N = 506) | Chi-square p-value |
| **Sex** | | | | | | |
| Female | 310 (43.1) | 203 (40.1) | 0.305 | 309 (42.9) | 203 (40.1) | 0.328 |
| Male | 410 (56.9) | 303 (59.9) | | 411 (57.1) | 303 (59.9) | |
| **Age group** | | | | | | |
| 15–24 | 142 (19.7) | 77 (15.2) | 0.154 | 78 (10.8) | 33 (6.5) | 0.036 |
| 25–34 | 310 (43.1) | 216 (42.7) | | 256 (35.6) | 196 (38.7) | |
| 35–44 | 237 (32.9) | 190 (37.5) | | 294 (40.8) | 199 (39.3) | |
| 45+ | 31 (4.3) | 23 (4.5) | | 92 (12.8) | 78 (15.4) | |
| **Marital status** | | | | | | |
| Married | 453 (62.9) | 316 (62.5) | 0.370 | 490 (68.1) | 334 (66) | 0.720 |
| Separated/Divorced | 182 (25.3) | 14 1(27.9) | | 179 (24.9) | 136 (26.9) | |
| Never married | 85 (11.8) | 49 (9.7) | | 51 (7.1) | 36 (7.1) | |
| **Occupation** | | | | | | |
| Agriculture | 49 (6.8) | 41(8.1) | 0.072 | 52 (7.2) | 37 (7.3) | 0.205 |
| Administrative | 14 (1.9) | 6 (1.2) | | 8 (1.1) | 4(0.8) | |
| Trading | 175 (24.3) | 100 (19.8) | | 173 (24) | 112 (22.1) | |
| Fishing | 226 (31.4) | 195(38.5) | | 224 (31.1) | 193 (38.1) | |
| Student | 16 (2.2) | 6 (1.2) | | 2 (0.3) | 0 (0.0) | |
| Unemployed | 64 (8.9) | 37 (7.3) | | 66 (9.2) | 38 (7.5) | |
| Others | 176 (24.4) | 121 (23.9) | | 195 (27.1) | 122 (24.1) | |

(n = 855) of pre-COVID-19 participants were lost to follow-up. The direct impacts of flooding and COVID-19 on primary outcomes were assessed in the remaining 1,226 participants. Out-migration and absence from work were the dominant reasons for lost to follow-up (S2 Table). Participants in both surveys and included in the final analysis significantly differed from those lost to follow-up on several factors, including sex, age, marital status, occupation, HIV risk score, and reported engagement in transactional sex (S3 Table).

Of the 1,226 participants in the pre- and post-COVID-19 surveys (Table 1), 506 were from households directly impacted by floods. Of those who were affected by floods, 40.1% (n = 203) were female and 66% (n = 334) were married at the post-COVID-19 survey. Demographic characteristics participants from households not directly impacted by floods were similar. Compared to the initial survey, participants in post-COVID-19 survey were somewhat older, which was expected because of the closed cohort study design.

Between the pre- and post-COVID-19 surveys, we observed somewhat lower levels of HIV risk behaviors and higher levels of HIV treatment coverage (Table 2). Significant declines were observed in inconsistent condom use with non-marital partners and self-reported transactional sex, irrespective of household flooding status (Table 3 and Fig 3). Relatedly, we observed a decline in HIV risk score among HIV-seronegative participants, regardless of flooding status (Table 2 and Fig 4). There was a statistically significant increase in ART uptake among flooded households (92.0% to 97.9%, p = 0.008) and non-flooded households (91.3% to 96.8%, p = 0.007). There was a non-statistically significant increase in HIV viral load suppression among flooded households (85.0% to 88%, p = 0.108) and non-flooded households (87.9% to 88.8%, p = 0.126).

We observed a 17% decline in transactional sex (aPR = 0.83, 95% CI: 0.75–0.92) and a 28% decline in the overall HIV risk score (aPR = 0.72, 95% CI: 0.64–0.80) among HIV-negatives

**Table 2. Comparison of sexual behaviors and HIV related outcomes in the pre- and post-COVID periods.**

| | Pre-COVID | Post-COVID | Chi-square p-value |
|---|---|---|---|
| | N = 1226 | N = 1226 | |
| **HIV risk group** | | | |
| 0–50 | 17 (1.4) | 67 (5.5) | <0.001 |
| 51–100 | 239 (19.5) | 330 (26.9) | |
| 101–150 | 499 (40.7) | 546 (44.5) | |
| 151–200 | 391 (31.9) | 241 (19.7) | |
| 201–250 | 68 (5.5) | 32 (2.6) | |
| 251+ | 12 (1) | 10 (0.8) | |
| **Number of sexual partners in the past 12 months** | | | |
| 0 to 1 partner | 732 (59.7) | 725 (59.1) | 0.773 |
| More than 1 partner | 494 (40.3) | 501 (40.9) | |
| **Inconsistent condom use with non-marital partner** | | | |
| No | 716 (58.4) | 772 (63) | 0.021 |
| Yes | 510 (41.6) | 454 (37) | |
| **Transactional sex** | | | |
| No | 866 (70.6) | 922 (75.2) | 0.011 |
| Yes | 360 (29.4) | 304 (24.8) | |
| **HIV status** | | | |
| Negative | 774 (63.1) | 757 (61.7) | 0.478 |
| Positive | 452 (36.9) | 469 (38.3) | |
| **ART status (among people living with HIV)** | | | |
| Not on ART | 38 (8.4) | 13 (2.8) | <0.001 |
| Yes | 414 (91.6) | 456 (97.2) | |
| **Viral load suppression (among people living with HIV)** | | | |
| Suppressed | 392 (87.3) | 415 (91.8) | 0.027 |
| Not suppressed | 57 (12.7) | 37 (8.2) | |

between the pre- and post-COVID periods. ART coverage increased by 5% (from 91.6% to 97.2%, p<0.001) among people living with HIV between the pre- and post-COVID periods (Table 4).

In difference-in-differences analyses, we found no significant changes in the differences in HIV risk behaviors, disease burden, or viral suppression outcomes when comparing the group affected by floods to those not affected by floods in the pre- and post-COVID time periods (Table 5 and S1 Fig). Results were similar in matched-pair conditional logistic regression analyses (S4 Table).

## Discussion

Despite the co-occurrence of flooding and the COVID-19 pandemic in a high HIV risk population, our analysis observed declines in transactional sex and HIV risk score among HIV-negatives as well as an increase in ART coverage. We observed no adverse impact of these extreme environmental conditions on HIV risk behaviors, seroprevalence, or virological outcomes among individuals who remained in the community.

These findings are consistent with findings from a study that assessed effects of COVID-19 on HIV services across eleven sub-Saharan African countries, which noted an increase in the number of people living with HIV who are on ART and virally suppressed during the COVID-19 pandemic [33]. In East Africa, despite significant societal and environment disruptions,

**Table 3. Comparison of HIV-associated behavioral characteristics of among 1,266 participants by household flooding status in the pre- and post-COVID-19 surveys.**

| | Flooded (N = 506) | | | Non-Flooded (N = 720) | | |
|---|---|---|---|---|---|---|
| HIV associated behaviors | Pre-COVID | Post-COVID | McNemar Test P value | Pre-COVID | Post-COVID | McNemar Test P value |
| **Number of sexual partners in the past 12 months** | | | | | | |
| 0 to 1 partner | 309 (61.1) | 305 (60.3) | 0.885 | 423 (58.8) | 420 (58.3) | 0.809 |
| More than 1 partner | 197 (38.9) | 201 (39.7) | | 297(41.3) | 300 (41.7) | |
| **Inconsistent condom use with non-marital partners** | | | | | | |
| No | 295 (58.3) | 320 (63.2) | 0.058 | 421 (58.5) | 452 (62.8) | 0.061 |
| Yes | 211 (41.7) | 186 (36.8) | | 299 (41.5) | 268 (37.2) | |
| **Transactional sex** | | | | | | |
| No | 365 (72.1) | 388 (76.7) | 0.004 | 501 (69.6) | 534 (74.2) | 0.026 |
| Yes | 141 (27.9) | 118 (23.3) | | 219 (30.4) | 186(25.8) | |
| **HIV status** | | | | | | |
| Negative | 319 (63.0) | 314 (62.1) | <0.001 | 455 (63.2) | 443 (61.5) | 0.062 |
| Positive | 187 (37.0) | 192 (37.9) | | 265 (36.8) | 277 (38.5) | |
| **Current ART use (among people living with HIV)** | | | | | | |
| Not on ART | 15 (8.0) | 4 (2.1) | <0.001 | 23 (8.7) | 9 (3.2) | <0.001 |
| On ART | 172 (92.0) | 188 (97.9) | | 242 (91.3) | 268 (96.8) | |
| **Viral load suppression (among people living with HIV)** | | | | | | |
| Suppressed | 159 (85.0) | 169 (88.0) | 0.007 | 233 (87.9) | 246 (88.8) | 0.024 |
| Not suppressed | 27 (14.4) | 17 (8.8) | | 30 (11.3) | 20 (7.3) | |
| Missing | 1 (0.5) | 6 (3.1) | | 2 (0.7) | 11 (3.9) | |

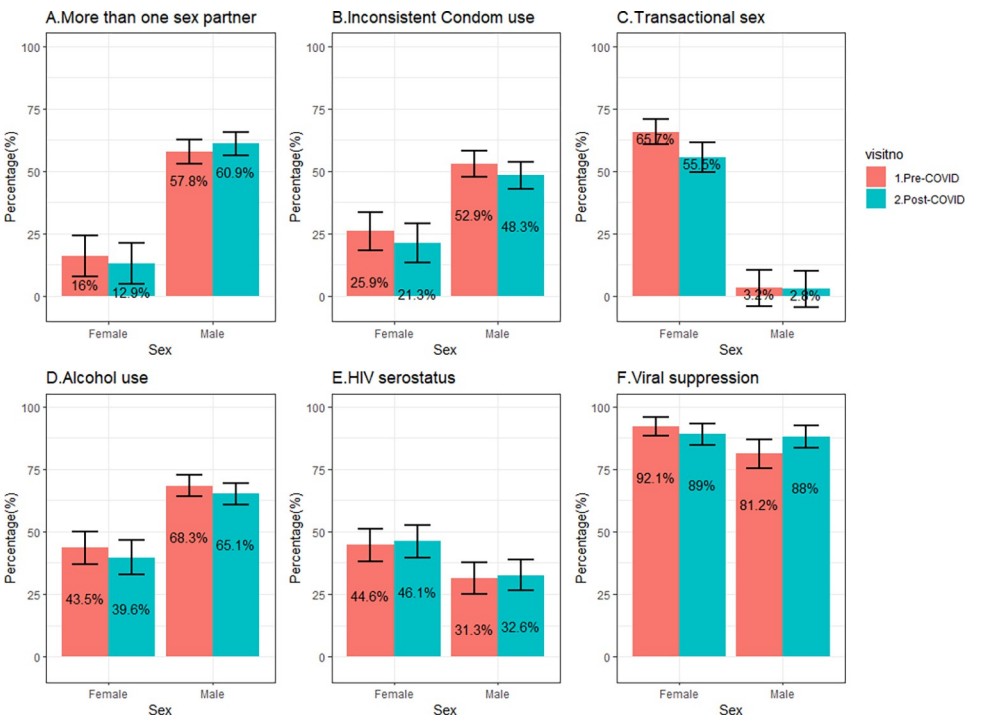

**Fig 3. Comparison of HIV risk behaviors and HIV outcomes by sex and COVID exposure.**

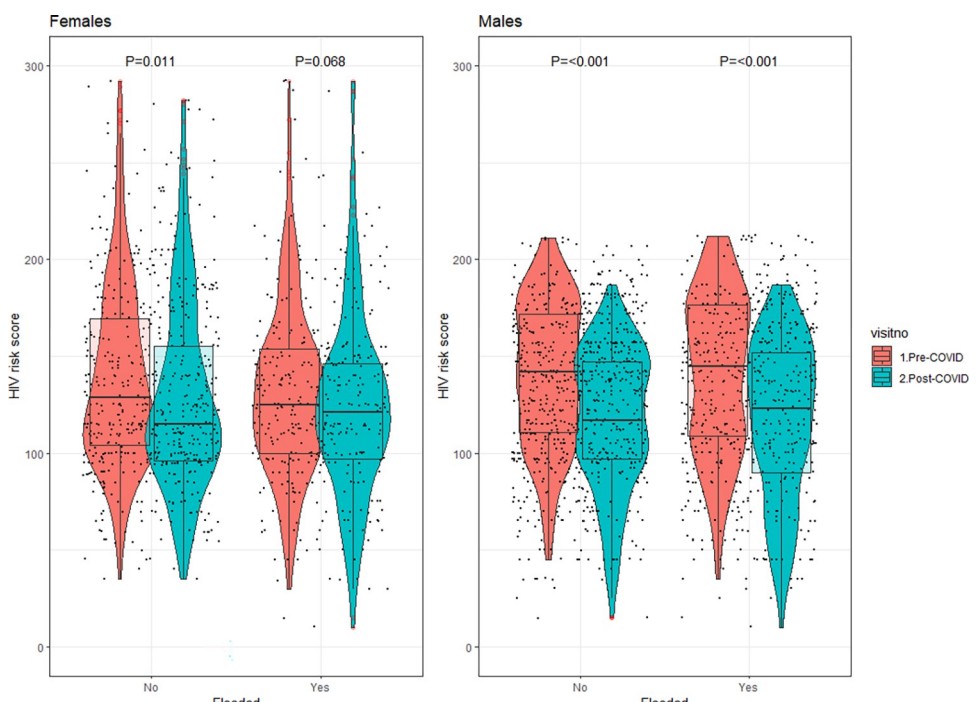

**Fig 4. Comparison of overall HIV risk scores among females and males, by exposure to flooding and COVID.**

**Table 4. Unadjusted and adjusted modified Poisson statistical model for HIV risk behaviors and outcomes in pre- and post-COVID-19 periods.**

| | PR (95% CI) | P value | Adjusted PR (95% CI) | P value |
|---|---|---|---|---|
| **Sexual partners in the 12 months** | | | | |
| Pre-COVID | 1 | | 1 | |
| Post-COVID | 1.01 (0.92–1.11) | 0.774 | 1.05 (0.96–1.14) | 0.299 |
| **Inconsistent condom use with non-marital partner** | | | | |
| Pre-COVID | 1 | | 1 | |
| Post-COVID | 0.89 (0.81–0.98) | 0.023 | 0.96 (0.88–1.06) | 0.438 |
| **Transactional sex** | | | | |
| Pre-COVID | 1 | | 1 | |
| Post-COVID | 0.84 (0.74–0.96) | 0.011 | 0.83 (0.75–0.92) | 0.001 |
| **HIV risk score** | | | | |
| Pre-COVID | 1 | | 1 | |
| Post-COVID | 0.60 (0.53–0.68) | <0.001 | 0.72 (0.64–0.80) | <0.001 |
| **HIV prevalence** | | | | |
| Pre-COVID | 1 | | 1 | |
| Post-COVID | 1.04 (0.94–1.15) | 0.479 | 0.93 (0.85–1.03) | 0.177 |
| **Current ART use (among people living with HIV)** | | | | |
| Pre-COVID | 1 | | 1 | |
| Post-COVID | 1.06 (1.03–1.10) | <0.001 | 1.05 (1.02–1.08) | 0.002 |
| **Viral load suppression (among people living with HIV)** | | | | |
| Pre-COVID | 1 | | 1 | |
| Post-COVID | 1.05 (1.01–1.10) | 0.027 | 1.03 (0.98–1.07) | 0.249 |

**Table 5. Unadjusted and adjusted difference-in-difference statistical model between flooded and non-flooded participants in pre- and post-COVID-19 period.**

| Outcomes | Unadjusted | | Adjusted | |
|---|---|---|---|---|
| | DID (95%CI) | p value | DID (95%CI) | p value |
| More than one sexual partner in the past 12 months | 1.13 (0.86–1.47) | 0.38 | 1.01 (0.86–1.17) | 0.92 |
| Inconsistent condom use with non-marital partner | 0.98 (0.83–1.16) | 0.85 | 0.98 (0.83–1.15) | 0.81 |
| Transactional sex | 0.98 (0.82–1.19) | 0.88 | 0.98 (0.81–1.18) | 0.84 |
| HIV prevalence | 0.98 (0.79–1.21) | 0.87 | 1.00 (0.95–1.04) | 0.94 |
| Current ART use (among people living with HIV) | 1.00 (0.94–1.07) | 0.88 | 1.01 (0.95–1.08) | 0.67 |
| Viral load suppression (among people living with HIV) | 1.02 (0.93–1.11) | 0.67 | 1.03 (0.95–1.12) | 0.41 |

DID: difference-in-differences statistical model

relatively few challenges to ART adherence were reported [34, 35]. In this hyperendemic fishing community, we observed declines in inconsistent condom use with non-marital sexual partners, transactional sex, and overall HIV risk score when comparing pre-COVID-19 and post-COVID-19 in both the flooded and non-flooded groups. It is possible that the existence of COVID-19 and related health messages during pandemic response increased overall health consciousness [36, 37]. Notably, a reduction in HIV risk factors was observed prior to the dual crises, consistent with what has been reported in this population between the pre- and post-COVID-19 periods [10].

It is not inconceivable that there was a decline in inconsistent condom use, given that there was a reduction in transactional sex as well. In times of adverse crisis of this nature, it could be that there was little to transact in, therefore transactional sex may not have been an option. Also, disposal income could have been affected so people could hardly engage in luxuries [38]. Although the opposite is possible too (i.e., people are more likely to engage in transactional sex in times of crisis), this is premised on the fact that there is one group that has material items and another that lacks material items. In the case of this community in early 2020, there is no reason to suspect that some group(s) had preferential access to items that could be used for transactions.

Resilience to flooding and COVID-19 among people living with HIV in this population may have been bolstered by the presence of supportive social networks. People living with HIV who have strong support systems, such as healthcare providers, community peer leaders and friends can rely on these networks for assistance, emotional support, and access to necessary resources during times of emergency [39–41]. In addition, this study focused on individuals who survived life-threatening trauma from flooding and COVID-19. Experiencing these traumas may lead individuals to have enhanced awareness of their vulnerability and the importance of maintaining good health and prioritizing healthcare services [42]. The HIV resilience observed in this population may also be attributed to innovative programs implemented during the COVID-19 pandemic, such as the use of motorcycle taxis, multi-month dispensing of ARVs, and ART delivery approaches (e.g. fast-track, home, peer, community pharmacy, and community client-led models) [25]. Addressing HIV program needs in the context of disasters has been shown to improve outcomes in similar settings [7]. The use of telemedicine, as employed in this setting, has also been reported to improve health outcomes on HIV risk indicators during COVID-19 [43].

This study had limitations. First, the simultaneous occurrence of flooding and COVID-made it challenging to assess the impact of each separately. Second, there was selection bias among respondents, as the study focused on individuals who remained in the community

both before and after flooding occurred; these participants differed from those who migrated out of the community (migration could potentially be due to being more affected by the flooding, or having more means to relocate). Thirdly, the community experienced other social disruptions: for example, the Ugandan army targeted communities which depended on subsistence fishing on Lake Victoria and other lakes to eliminate what the state considered illegal fishing after a 2020 update in the national fisheries and aquaculture policy [44]. These and other unmeasured activities in fishing communities may make it difficult to control for all confounding factors in assessing the effect of flooding and COVID-19 on HIV risk behaviors.

In conclusion, despite a high background burden of HIV, the COVID-19 pandemic, and serious flooding, we observed no adverse impact on HIV risk behaviors, burden, or virological outcomes in this HIV hyperendemic fishing community. These findings highlight the importance of integrating HIV programming into disaster response efforts, which can inform customized interventions to lessen the impact of natural disasters on HIV. The resilience observed in HIV outcomes during crises underscores the significance of adaptable health systems, providing valuable insights to reinforce resilience in resource-limited settings. Innovations such as telemedicine and community-led ART delivery are pivotal in sustaining HIV care during emergencies. By incorporating these strategies into regular programming, access to services can be improved, especially for hard-to-reach populations. Understanding exactly what HIV programs and personal/community-level strategies worked to maintain good public health outcomes despite extreme environmental and pandemic conditions may help improve population health.

## Supporting information

**S1 Table. Summary of participation and non-participation in the pre- and post-COVID periods.**
(DOCX)

**S2 Table. Reasons for lost to follow-up in the pre- and post-COVID periods.**
(DOCX)

**S3 Table. Comparison of baseline characteristics of respondents who were and were not lost to follow-up between the two RCCS surveys.**
(DOCX)

**S4 Table. Matched-pair conditional logistic regression analyses.**
(DOCX)

**S1 Fig. Spaghetti plot for HIV risk score, by gender, comparing flooding and COVID exposure.**
(DOCX)

## Acknowledgments

We thank the Rakai Community Cohort Study participants and the Rakai Health Sciences Program team for making this study possible.

## Author Contributions

**Conceptualization:** Hadijja Nakawooya, Victor Ssempijja, Larry W. Chang, Joseph Kagaayi, M. Kate Grabowski, Robert Ssekubugu.

**Data curation:** Hadijja Nakawooya, Anthony Ndyanabo.

**Formal analysis:** Hadijja Nakawooya, Victor Ssempijja, Anthony Ndyanabo.

**Investigation:** Ping Teresa Yeh, Robert Ssekubugu.

**Methodology:** Hadijja Nakawooya, M. Kate Grabowski.

**Visualization:** Ping Teresa Yeh.

**Writing – original draft:** Hadijja Nakawooya, Ping Teresa Yeh, Larry W. Chang, M. Kate Grabowski.

**Writing – review & editing:** Hadijja Nakawooya, Victor Ssempijja, Anthony Ndyanabo, Ping Teresa Yeh, Larry W. Chang, Maria J. Wawer, Fred Nalugoda, David Serwadda, Ronald H. Gray, Steven J. Reynolds, Tom Lutalo, Godfrey Kigozi, M. Kate Grabowski, Robert Ssekubugu.

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
