## [Decision Letter · Decision Letter 0]

29 Feb 2024

PONE-D-23-33428

Impact of natural disasters on HIV risk behaviors, seroprevalence, and virological supression in a hyperendemic fishing village in Uganda

PLOS ONE

Dear Dr. NAKAWOOYA, 

Thank you for submitting your manuscript to PLOS ONE. After careful consideration, we feel that it has merit but does not fully meet PLOS ONE’s publication criteria as it currently stands. Therefore, we invite you to submit a revised version of the manuscript that addresses the points raised during the review process.

The revised version should explain the rational or hypothesis for studying flooding and COVID-19 for HIV in the introduction and elucidate the public health implications of this study. Additionally, the revision ought to address all the statistical issues raised by the reviewer, ensure coherence between tables/figures and the narrative, and enhance the overall readability of the manuscript. 

We look forward to receiving your revised manuscript.

Kind regards,

Xiaoyu Song

Academic Editor

PLOS ONE

Journal Requirements:

4. In this instance it seems there may be acceptable restrictions in place that prevent the public sharing of your minimal data. However, in line with our goal of ensuring long-term data availability to all interested researchers, PLOS’ Data Policy states that authors cannot be the sole named individuals responsible for ensuring data access (http://journals.plos.org/plosone/s/data-availability#loc-acceptable-data-sharing-methods).

6. We note that Figure 2 in your submission contain map/satellite images which may be copyrighted. All PLOS content is published under the Creative Commons Attribution License (CC BY 4.0), which means that the manuscript, images, and Supporting Information files will be freely available online, and any third party is permitted to access, download, copy, distribute, and use these materials in any way, even commercially, with proper attribution. For these reasons, we cannot publish previously copyrighted maps or satellite images created using proprietary data, such as Google software (Google Maps, Street View, and Earth). For more information, see our copyright guidelines: http://journals.plos.org/plosone/s/licenses-and-copyright.

Reviewers' comments:

Reviewer's Responses to Questions

**Comments to the Author**

1. Is the manuscript technically sound, and do the data support the conclusions?

Reviewer #1: Yes

2. Has the statistical analysis been performed appropriately and rigorously? 

Reviewer #1: Yes

3. Have the authors made all data underlying the findings in their manuscript fully available?

Reviewer #1: No

4. Is the manuscript presented in an intelligible fashion and written in standard English?

Reviewer #1: Yes

5. Review Comments to the Author

Reviewer #1: The manuscript used the data from an open longitudinal population-based HIV surveillance cohort in Lake Victoria fishing communities, which were affected by severe flooding and COVID-19 at the same time, to assess the changes in HIV-associated behaviors and treatment outcomes before and after COVID-19 among flooded and non-flooded households. To address the research questions, a difference-in-differences analysis was employed using a modified Poisson regression model.

While the manuscript generally succeeds in describing the research questions, conducting statistical analyses, and presenting results adequately, there are still a lot of improvements needed for publication purposes. Specifically, there is a need for a more detailed elaboration on the background and purpose of the research study. Consideration could be given to employing a better model which accounts for the within-subject correlation. Additionally, several suggestions for correcting errors in the results section have been provided in the comments below.

Major comments:

1. The author should provide further justifications for the research objective and the reason of choosing population in Lake Victoria fishing communities as study population. While it is intuitive to recognize the importance of studying the impacts of floods or COVID lockdown on HIV separately, readers may not immediately grasp why understanding the integrated effects of flooding and COVID-19 on HIV is essential.

2. While the modified Poisson regression model is suitable for addressing the research questions, the author should offer justifications for choosing this model over other commonly used models, such as the log-binomial model. Furthermore, it is essential for the author to cite the method paper of the modified Poisson regression to provide proper reference in the statistical analysis section.

3. Since the samples can be perfectly matched in the pre- and post-COVID-19 surveys, the author can consider using DID model with random effects or generalized estimating equation (GEE) to account for the within-subject correlation.

4. The manuscript needs editing for language and writing quality. The language is sometimes difficult to follow. Additionally, there are several typos in the manuscripts. One of the major ones is “difference in difference analysis” should be “difference-in-differences analysis”. The author should correct the typos and revise the language to improve readability, especially for the introduction section.

Minor comments:

1. The author should employ clearer language when describing the primary outcomes in the analysis, ensuring that readers can easily distinguish the variable type, such as binary or count data. For instance, in the first sentence of the primary outcomes section, changing "inconsistent condom use" to "the frequency of inconsistent condom use" or "whether inconsistent condom use occurs" would clarify that it is a binary or count data, aiding readers in better understanding.

2. Since supplementary table 4 was mentioned many times in the results section, the author should consider moving the table to the main text instead of placing it in the supplementary information.

3. In figure 3, the error bars are not fully represented in sub-figures C and F. For figure 3 and 4, there seems to be no clear rationale for stratifying by gender, especially considering that gender is not a variable of interest, and there is no significant difference in gender distribution as indicated in Table 1a.

4. In the results section, it was said “we observed significant declines in inconsistent condom use with non-marital partners and self-reported transactional sex, irrespective of household flooding status (Table 1b, Figure 3)”. Table 1b and figure 3 are not related to the conclusion.

5. In table 2 and supplementary table 5, “1” should be changed to “Reference”.

6. In the last paragraph of results section, the interpretations of DID model results are not accurate. “No significant differences” should be changed to “no significant changes in differences”.

7. The author is aware that there are significant differences between participants included in the final analysis and participants lost to follow-up as indicated in supplementary table 3 but did not mention that the selection method utilized in the study could cause selection bias in the limitation of study.

8. The author should elaborate more on how the results of the study will help inform public health policies and improve population health in the future in the discussion section.

6. PLOS authors have the option to publish the peer review history of their article (what does this mean?). If published, this will include your full peer review and any attached files.

Reviewer #1: **Yes: **Weijia Fu

---

## [Author Response · Author response to Decision Letter 0]

11 Jun 2024

May 24, 2024

To Xiaoyu Song, Academic Editor, 

Thank you for the positive review of our manuscript, “Impact of natural disasters on HIV risk behaviors, seroprevalence, and virological supression in a hyperendemic fishing village in Uganda” [PONE-D-23-33428] for consideration by PLOS ONE. 

Below, we have included the comments from the reviewer and the academic editor and how we have responded to each in blue. We appreciate the comments from both reviewers and believe they have strengthened the manuscript. We hope that these revisions address the points raised during the review process and that you will accept this manuscript for publication.

Sincerely,

Hadijja Nakawooya

Statistician

Rakai Health Sciences Program

Kalisizo, Uganda

Reviewer 1

The manuscript used the data from an open longitudinal population-based HIV surveillance cohort in Lake Victoria fishing communities, which were affected by severe flooding and COVID-19 at the same time, to assess the changes in HIV-associated behaviors and treatment outcomes before and after COVID-19 among flooded and non-flooded households. To address the research questions, a difference-in-differences analysis was employed using a modified Poisson regression model. While the manuscript generally succeeds in describing the research questions, conducting statistical analyses, and presenting results adequately, there are still a lot of improvements needed for publication purposes. Specifically, there is a need for a more detailed elaboration on the background and purpose of the research study. Consideration could be given to employing a better model which accounts for the within-subject correlation. Additionally, several suggestions for correcting errors in the results section have been provided in the comments below.

Thank you for this generally positive review.

Major comments:

1. The author should provide further justifications for the research objective and the reason of choosing population in Lake Victoria fishing communities as study population. While it is intuitive to recognize the importance of studying the impacts of floods or COVID lockdown on HIV separately, readers may not immediately grasp why understanding the integrated effects of flooding and COVID-19 on HIV is essential.

We appreciate the reviewer's valuable input. As per their suggestion, we have made additions and modifications to the sentences in the first paragraph of the introduction, elucidating the rationale behind selecting the population from Lake Victoria fishing communities as our study sample: “Lake Victoria fishing communities in eastern Africa have one of the highest HIV burdens globally, with adult HIV seroprevalence estimates ranging from 20 to 40% and HIV transmission remains high with incidence of 1.59/100 per person years [10, 11]. Many people in this fishing community engage in high-risk sexual behaviors and drug and alcohol use. Women frequently report risky sex behaviors which is often related to sex and bar work. The situation is further complicated with increased vulnerability to HIV, including increased peer pressure especially among adolescents and poor housing comprised of make-shift shelters made of timber and occupied by two or more families of different backgrounds and their children.”

We have also made additions and modifications to the sentences in the second paragraph of the introduction, elucidating the rationale of studying the impacts of floods or COVID lockdown on HIV separately: “Floods disrupt lives and health practices, raising HIV transmission risks due to behavior changes and limited prevention resources[9, 17]. Reduced support for individuals with HIV, coupled with limited information on HIV and the disruption of counseling and testing services by floods, contributes to undiagnosed cases and unsafe sex practices[18]. Malnutrition and disease weaken immune systems, increasing virus transmission likelihood[7, 19]”..”

2. While the modified Poisson regression model is suitable for addressing the research questions, the author should offer justifications for choosing this model over other commonly used models, such as the log-binomial model. Furthermore, it is essential for the author to cite the method paper of the modified Poisson regression to provide proper reference in the statistical analysis section.

We thank the reviewer for this important comment. Two citations have been added to support the use of the modified Poisson regression in the statistical analysis section:

Chen, W., et al., Comparison of robustness to outliers between robust poisson models and log-binomial models when estimating relative risks for common binary outcomes: a simulation study. BMC Medical Research Methodology, 2014. 14(1): p. 82.

Gallis, J.A. and E.L. Turner, Relative Measures of Association for Binary Outcomes: Challenges and Recommendations for the Global Health Researcher. Ann Glob Health, 2019. 85(1): p. 137

3. Since the samples can be perfectly matched in the pre- and post-COVID-19 surveys, the author can consider using DID model with random effects or generalized estimating equation (GEE) to account for the within-subject correlation.

Thank for this excellent suggestion. We have now used a DID model with random effects applied to participant id to account for the within-subject correlation and updated our methods section (“The assessment of difference-in-differences (DID) was conducted by examining the coefficient of the interaction term between these variables, with random effects analysis applied to participant ID to account for the within-subject correlation.”) and results accordingly.

4. The manuscript needs editing for language and writing quality. The language is sometimes difficult to follow. Additionally, there are several typos in the manuscripts. One of the major ones is “difference in difference analysis” should be “difference-in-differences analysis”. The author should correct the typos and revise the language to improve readability, especially for the introduction section.

Thank you for catching this. We have adjusted our language to say “difference-in-differences analysis” and have gone through the entire manuscript again for readability.

Minor comments:

1. The author should employ clearer language when describing the primary outcomes in the analysis, ensuring that readers can easily distinguish the variable type, such as binary or count data. For instance, in the first sentence of the primary outcomes section, changing "inconsistent condom use" to "the frequency of inconsistent condom use" or "whether inconsistent condom use occurs" would clarify that it is a binary or count data, aiding readers in better understanding.

Thank you for catching this. We have edited our phrasing of the condom use outcome to say "whether inconsistent condom use occurs", so that it is clear to readers that this is a binary variable.

2. Since supplementary table 4 was mentioned many times in the results section, the author should consider moving the table to the main text instead of placing it in the supplementary information.

Thank you for this feedback. Supplementary Table 4 has been relocated to the main text and renamed as Table 2, and we have renumbered the subsequent tables.

3. In figure 3, the error bars are not fully represented in sub-figures C and F. For figure 3 and 4, there seems to be no clear rationale for stratifying by gender, especially considering that gender is not a variable of interest, and there is no significant difference in gender distribution as indicated in Table 1a.

Thank you for noticing this – we have edited the figure to ensure that the error bars are fully presented in each of the six plots. We chose to stratify by gender because men and women in this community have historically demonstrated different risk behaviors and reported different HIV outcomes.

4. In the results section, it was said “we observed significant declines in inconsistent condom use with non-marital partners and self-reported transactional sex, irrespective of household flooding status (Table 1b, Figure 3)”. Table 1b and figure 3 are not related to the conclusion.

Thank you for the catch; we had switched the table and figure – we have adjusted this to cite figure 3 and (what is now) table 3.

5. In table 2 and supplementary table 5, “1” should be changed to “Reference”.

We have made this change.

6. In the last paragraph of results section, the interpretations of DID model results are not accurate. “No significant differences” should be changed to “no significant changes in differences”.

We have made this change.

7. The author is aware that there are significant differences between participants included in the final analysis and participants lost to follow-up as indicated in supplementary table 3 but did not mention that the selection method utilized in the study could cause selection bias in the limitation of study.

Thank you for this comment. We have added this to the study limitations “Second, there was selection bias among respondents, as the study focused on individuals who stayed in the community both before and after flooding occurred; these participants differed from those who migrated out of the community, potentially due to being more affected by the flooding, thus leading to their migration.”

8. The author should elaborate more on how the results of the study will help inform public health policies and improve population health in the future in the discussion section.

We have added elaboration on the public health implications of this study to the discussion section: “In conclusion, despite a high background burden of HIV, the COVID-19 pandemic, and serious flooding, we observed no adverse impact on HIV risk behaviors, burden, or virological outcomes in this HIV hyperendemic fishing community. These findings highlights the importance of integrating HIV programming into disaster response efforts, which can inform customized interventions to lessen the impact of natural disasters on HIV. The resilience observed in HIV outcomes during crises underscores the significance of adaptable health systems, providing valuable insights to reinforce resilience in resource-limited settings. Innovations such as telemedicine and community-led ART delivery are pivotal in sustaining HIV care during emergencies. By incorporating these strategies into regular programming, access to services can be improved, especially for hard-to-reach populations. Understanding exactly what HIV programs and personal/community-level strategies worked to maintain good public health outcomes despite extreme environmental and pandemic conditions may help improve population health.”

Editor

The revised version should explain the rational or hypothesis for studying flooding and COVID-19 for HIV in the introduction and elucidate the public health implications of this study. Additionally, the revision ought to address all the statistical issues raised by the reviewer, ensure coherence between tables/figures and the narrative, and enhance the overall readability of the manuscript.

Thank you for these overarching comments. We have added the rationale for examining the impact of COVID and flooding on HIV to the introduction, and have elucidated the public health implications of this study in the discussion section. We have also addressed the statistical issues raised by the reviewer (see responses above), ensured coherence between the tables/figures and the narrative, and doubled checked the overall readability of the manuscript.

Journal Requirements:

We have doubled-checked style requirements per the style templates.

We have completed this questionnaire and uploaded it as supporting information.

We have removed all funding-related text in the manuscript.

4. In this instance it seems there may be acceptable restrictions in place that prevent the public sharing of your minimal data. However, in line with our goal of ensuring long-term data availability to all interested researchers, PLOS’ Data Policy states that authors cannot be the sole named individuals responsible for ensuring data access (http://journals.plos.org/plosone/s/data-availability#loc-acceptable-data-sharing-methods).

We appreciate the effort that the PLOS journal family takes to make data available (and to guarantee long-term stability and availability of data despite potential changes to the institutional affiliations etc of individual authors). Data requests may be sent to the Rakai Health Sciences Program data management office (datarequests@rhsp.org), where data are archived across all the various projects run by the RHSP (original paper forms from the RCCS surveys, as well as the electronic datasets for each survey round).

We have inserted the ORCID iD for the corresponding author: 0000-0002-0566-547X

6. We note that Figure 2 in your submission contain map/satellite images which may be copyrighted. All PLOS content is published und

---

## [Decision Letter · Decision Letter 1]

24 Jul 2024

PONE-D-23-33428R1Impact of natural disasters on HIV risk behaviors, seroprevalence, and virological supression in a hyperendemic fishing village in UgandaPLOS ONE

Dear Dr. NAKAWOOYA,

Thank you for submitting your manuscript to PLOS ONE. After careful consideration, we feel that it has merit but does not fully meet PLOS ONE’s publication criteria as it currently stands. Therefore, we invite you to submit a revised version of the manuscript that addresses the points raised during the review process.In the revised manuscript, please improve the readability and address the typos, such as  "we conducted and pre and post study to evaluate ..."  in the fourth paragraph, for its acceptance. 

We look forward to receiving your revised manuscript.

Kind regards,

Xiaoyu Song

Academic Editor

PLOS ONE

Journal Requirements:

Reviewers' comments:

Reviewer's Responses to Questions

**Comments to the Author**

1. If the authors have adequately addressed your comments raised in a previous round of review and you feel that this manuscript is now acceptable for publication, you may indicate that here to bypass the “Comments to the Author” section, enter your conflict of interest statement in the “Confidential to Editor” section, and submit your "Accept" recommendation.

Reviewer #1: (No Response)

2. Is the manuscript technically sound, and do the data support the conclusions?

Reviewer #1: Yes

3. Has the statistical analysis been performed appropriately and rigorously? 

Reviewer #1: Yes

4. Have the authors made all data underlying the findings in their manuscript fully available?

Reviewer #1: (No Response)

5. Is the manuscript presented in an intelligible fashion and written in standard English?

Reviewer #1: (No Response)

6. Review Comments to the Author

Reviewer #1: The manuscript utilized data from Lake Victoria fishing communities, which were simultaneously affected by severe flooding and COVID-19, to assess changes in HIV-associated behaviors and treatment outcomes before and after COVID-19 among flooded and non-flooded households. In the initial submission, requests were made for a more detailed elaboration on the rationale and purpose of the study, as well as corrections to the results section.

Major revisions:

1. In the introduction section of the manuscript, the author elaborated more on the rationale for using data from fishing communities and studying the impacts of floods on HIV. This addition addressed the reviewers' and editors' requests for a more detailed explanation of the rationale or hypothesis behind investigating the relationship between flooding and HIV.

2. For statistical modeling, the DID model now included random effects to account for the within-subject correlation. Additionally, the author included two references for the statistical model used and revised the wording of the variables in the analysis to clarify their types.

3. In the discussion section, potential selection bias was addressed as one of the limitations of the study. Additionally, further elaborations on the public health implications of the study were included.

Minor revisions:

1. In the results section, the author addressed the issues pointed out by the reviewer and made changes/corrections accordingly, such as changing the phrasing of variables, editing the figures and moving table S4 to main text.

In general, the re-submitted manuscript addressed the issues pointed out by the reviewer and editor properly. However, the readability can still be improved by refining the wording, improving the logical flow, and considering restructuring parts of the manuscript. Additionally, there are still typos in the re-submitted manuscript, for example, the first sentence of the fourth paragraph in the introduction section: “we conducted and pre and post study to evaluate the impact of COVID-19 and flooding on …”.

Addressing these issues will help ensure that your valuable research is communicated as effectively as possible to readers and meet the journal's standards.

7. PLOS authors have the option to publish the peer review history of their article (what does this mean?). If published, this will include your full peer review and any attached files.

Reviewer #1: **Yes: **Weijia Fu

---

## [Author Response · Author response to Decision Letter 1]

5 Sep 2024

4. Have the authors made all data underlying the findings in their manuscript fully available?

Reviewer #1: (No Response)

Yes the data is available after completing a data request form.

5. Is the manuscript presented in an intelligible fashion and written in standard English?

Reviewer #1: (No Response) 

All the typographical or grammatical errors have been corrected.

---

## [Editor Report · Decision Letter 2]

22 Sep 2024

Impact of natural disasters on HIV risk behaviors, seroprevalence, and virological supression in a hyperendemic fishing village in Uganda

PONE-D-23-33428R2

Dear Dr. NAKAWOOYA,

We’re pleased to inform you that your manuscript has been judged scientifically suitable for publication and will be formally accepted for publication once it meets all outstanding technical requirements.

Kind regards,

Xiaoyu Song

Academic Editor

PLOS ONE
---

## [Editor Report · Acceptance letter]

3 Oct 2024

PONE-D-23-33428R2 

PLOS ONE

Dear Dr. NAKAWOOYA, 

I'm pleased to inform you that your manuscript has been deemed suitable for publication in PLOS ONE. Congratulations! Your manuscript is now being handed over to our production team.

Kind regards, 

on behalf of

Dr. Xiaoyu Song 

Academic Editor

PLOS ONE